# Estimation of (Co) Variance Components and Genetic Parameters for Pre- and Post-Weaning Growth Traits in Dağlıç Sheep

**DOI:** 10.3390/ani14010108

**Published:** 2023-12-27

**Authors:** Serdar Koçak, Samet Çinkaya, Mustafa Tekerli, Mustafa Demirtaş, Zehra Bozkurt, Koray Çelikeloğlu, Özlem Hacan, Metin Erdoğan

**Affiliations:** 1Department of Animal Science, Faculty of Veterinary Medicine, Afyon Kocatepe University, Afyonkarahisar 03200, Türkiye; tekerli@aku.edu.tr (M.T.); mustafademirtas@aku.edu.tr (M.D.); akinci@aku.edu.tr (Z.B.); kcelikeloglu@aku.edu.tr (K.Ç.); gucuyener@aku.edu.tr (Ö.H.); 2Department of Veterinary Biology and Genetics, Faculty of Veterinary Medicine, Afyon Kocatepe University, Afyonkarahisar 03200, Türkiye; erdogan@aku.edu.tr

**Keywords:** heritability, maternal effect, animal models, growth traits, Dağlıç sheep

## Abstract

**Simple Summary:**

The Dağlıç breed of sheep is fat-tailed, dual-purpose, and resistant to harsh environmental conditions in Türkiye. This research aimed to estimate (co) variance components and genetic parameters for pre- and post-weaning growth traits in Dağlıç sheep, considering the direct additive genetic, maternal genetic, and maternal permanent environmental effects with different statistical models. This study revealed moderate heritabilities for pre- and post-weaning growth variables of Dağlıç sheep. The genetic variation identified in this study suggests that selective breeding could yield favorable outcomes in the early growth of Dağlıç sheep in Türkiye.

**Abstract:**

The goal of this study was to estimate (co) variance components and genetic parameters for pre- and post-weaning growth traits in Dağlıç sheep, considering the direct additive genetic, maternal genetic, and maternal permanent environmental effects, with different statistical models. The information of 21,735 native Dağlıç lambs born between 2011 and 2021 was used to estimate (co) variance components by the Average Information-Restricted Maximum Likelihood algorithm. The results showed that the most suitable model was Model 3 for birth weight (BW), average daily gain (ADG), and weaning weight (WW). Model 4 was the most appropriate for weight at three (W3), weight at six (W6), and weight at twelve months of age (W12). The direct heritabilities for BW, W3, ADG, WW, W6, and W12 were 0.35 ± 0.02, 0.36 ± 0.03, 0.27 ± 0.02, 0.22 ± 0.02, 0.47 ± 0.05, and 0.47 ± 0.05, respectively. Genetic and phenotypic correlations amongst the traits were in the range of 0.103 ± 0.008 to 0.995 ± 0.002. These results can be used for the improvement of growth traits in the Dağlıç breed of sheep through selection.

## 1. Introduction

Türkiye is one of the major producers of sheep with a population of 45 million heads in 2021 [1]. Sheep are generally reared on smallholder farms in Türkiye. The main purposes of sheep husbandry are to benefit from the poor pastures and reduce the deficiency of meat products in the developing world. The growth of animals reflects adaptability to the environment, and high performance in growth traits affects the income of breeders [2]. Growth is influenced by both genetic and environmental effects [3]. Heritabilities and correlations among economically important growth traits are essential to the breeding programs for genetic improvement [4]. Ignoring the maternal and permanent environmental effects and covariance between offspring and dam may affect the heritability estimates [2,5]. Willham [6] and Meyer [7] also reported that the direct additive, maternal additive, and maternal permanent environmental variances should be considered when estimating a lamb’s genetic merit.

Dağlıç is one of the native breeds of sheep in Türkiye. The breed is fat-tailed, dual-purpose, resistant to harsh environmental conditions, and requires less feed and water than other breeds of sheep in Türkiye. The breeding area of the breed is the poor pastures of inner west Anatolia [8,9]. A community-based improvement program coordinated by General Directorate of Agricultural Research and Policies was started in 2011 to develop the growth traits of Dağlıç sheep. The heritabilities of these traits were previously estimated with the paternal half-sib method [10,11,12]. However, the effects that contribute to genetic variation for this breed remain unknown.

The aim of this study was to estimate (co) variance components and genetic parameters for pre- and post-weaning growth traits in native Dağlıç sheep considering the effects of direct additive genetic, maternal genetic, and maternal permanent environmental effects and correlation of dam and offspring with different models.

## 2. Materials and Methods

The study was conducted using the weight records and pedigree information of 21,735 native Dağlıç lambs born between 2011 and 2021. These animals were registered to the Afyonkarahisar sub-project (TAGEM/03DAG2011-01) of the “Community-based animal improvement program” project coordinated by the General Directorate of the Agricultural Research and Policies of the Ministry of Agriculture and Forestry of Türkiye. Animals used in this research were reared in Kurucaova village with the same environmental conditions. Kurucaova village is in the Bolvadin district of the Afyonkarahisar province, Türkiye. Kurucaova is located 1276 m above sea level at the 38°44′22.1″ N and 30°55′42.0″ E coordinates in western Anatolia. Semi-arid climate conditions prevail in the region. The breeders generally feed the animals with wheat or barley straw and a small amount of barley in winter. The mothers of the lambs used in the study are grazed on the natural pasture and stubble for 8 months of the year routinely. The lambs were weighed within 24 h after birth and raised with their mothers until weaning. Average weaning age was 124.89 days in this research. The weights of the lambs were taken consecutively until the lambs were older than one year. W3, W6, and W12 were estimated with interpolation using the weight records. The ADG was calculated from birth to weaning.

REML can use the fixed and random effects incorporated with pedigree information to estimate the genetic value of an animal with a single observation. Therefore, variance and covariance components for pre- and post-weaning growth traits were estimated using the Average Information-Restricted Maximum Likelihood (AI-REML) algorithm by WOMBAT software [13] considering a single-trait animal model. Only winter-born and single-birth lambs were used in the analysis. The Least-Squares analyses revealed that the fixed effects of year of birth, month of birth, sex, and dam age were significant for all traits. Additionally, the weaning age was also significant in the post-weaning growth traits. The farm was not considered as a fixed effect in the model because the conditions on the various farms were similar regarding animal housing, feeding, and management. Six single-trait animal models considering significant fixed effects with different combinations of additive genetic, maternal genetic, and maternal permanent environmental effects were used to estimate genetic parameters for each trait;
(1)$$Y=X\beta +Z_{a}a+e$$
(2)$$Y=X\beta +Z_{a}a+Z_{c}c+e$$
(3)$$Y=X\beta +Z_{a}a+Z_{m}m+e\;\;Cov\left(a,m\right)=0$$
(4)$$Y=X\beta +Z_{a}a+Z_{m}m+e\;\;Cov\left(a,m\right)=A\sigma_{am}$$
(5)$$Y=X\beta +Z_{a}a+Z_{m}m+Z_{c}c+e\;\;Cov\left(a,m\right)=0$$
(6)$$Y=X\beta +Z_{a}a+Z_{m}m+Z_{c}c+e\;\;Cov\left(a,m\right)=A\sigma_{am}$$
where Y is the vector of observations, β is the vector of fixed effects, a is the vector of direct additive genetic effects, m is the vector of maternal additive genetic effects, c is the vector of maternal permanent environmental effects, and e is the vector of residuals. X, Z*_a_*, Z*_m_*, and Z*_c_* are the incidence matrices related to the fixed, direct additive genetic, maternal additive genetic, and maternal permanent environmental effects, respectively. It was assumed that direct additive genetic, maternal additive genetic, maternal permanent environmental, and residual effects are normally and independently distributed with mean 0 and variance $A\sigma_{a}^{2}$, $A\sigma_{m}^{2}$, $I_{d}\sigma_{pe}^{2}$ and $I_{n}\sigma_{e}^{2}$, respectively. *A* is the numerator relationship matrix between animals, $\sigma_{a}^{2}$ is the direct additive genetic, $\sigma_{m}^{2}$ is the maternal additive genetic variance, $\sigma_{am}$ is the covariance between direct additive and maternal genetic effects, and $\sigma_{pe}^{2}$ and $\sigma_{e}^{2}$ are the maternal permanent environmental and residual variances, respectively. $I_{d}$ and $I_{n}$ are the identity matrices with orders equal to the number of dams and number of lambs, respectively. The log-likelihood ratio test was used to determine the most suitable model for each trait considering Akaike’s information criterion (AIC). The formula of AIC is as follows [14]:$$AIC=-2logL_{i}+2p_{i}$$
where log $L_{i}$ is the maximized log likelihood of model *i* at convergence, and $p_{i}$ is the number of random (co) variance parameters of model *i*. The model yielding the smallest AIC value was considered the most appropriate model.

The total heritability ($h_{T}^{2}$) for each model was calculated as $h_{T}^{2}=\left(\sigma_{a}^{2}+{0.5\sigma }_{m}^{2}+1.5\sigma_{am}\right)/\sigma_{P}^{2}$, which was described by Willham [6]. Genetic, phenotypic and residual correlations between pre- and post-weaning growth traits were estimated using a bivariate animal model (Model 1) in Wombat incorporating fixed effects.

## 3. Results and Discussion

The number of records and descriptive statistics describing the data structure used in the present study are shown in Table 1.

Estimated (co) variance components and genetic parameters together with LogL and AIC values for each model and trait are presented in Table 2 and Table 3. The model that had the lowest AIC value was chosen as the most suitable model. In the analysis of birth weight, AIREML did not converge for the models that contained the covariance between the direct additive and maternal genetic effects (models 4 and 6). The most suitable model for birth weight (BW), average daily gain (ADG), and weaning weight (WW) was Model 3, including direct and maternal effects. Model 4, including direct and maternal effects and covariances between them, was the most appropriate for weight at three (W3), weight at six (W6), and weight at twelve months of age (W12). The highest direct heritabilities for all traits were obtained in Model 1, which was not better than the other models according to the AIC values. 

Sharif et al. [15] revealed Model 3 as the most appropriate model for the analysis of birth weight of Lohi sheep, and this finding is in agreement with our study. Hızlı et al. [16] estimated the highest direct heritability with Model 1, and it was the most appropriate model for BW in Awassi sheep. On the contrary, Model 6 in Bonga [17] and Model 5 in Chokla sheep [18] were the most proper models. The direct heritability for birth weight was 0.35 ± 0.02 in Model 3. A similar estimate (0.34) was obtained by Oyieng et al. [19] in Red Maasai sheep. Conversely, our finding was higher than the values of 0.08 and 0.18 reported by Yalçın [11] and Ulusan and Bekyürek [12] using the sire model in Dağlıç sheep. Hızlı et al. [16] found a lower estimate of heritability (0.23) with Model 3 in Awassi sheep. Sharif et al. [15] and Balasundaram et al. [2] found low heritability estimates (0.15) for BW in Lohi and Mecheri sheep. The estimated maternal and total heritabilities for BW were 0.08 and 0.39, respectively, in our study. While the maternal heritability was low, the maternal genetic effect was significant for BW.

Model 4 was detected as the most suitable for W3 according to the values of AIC and LogL. Several researchers revealed that Models 1, 2, and 3 were the best for different sheep breeds [2,20,21,22]. The direct heritability estimate was 0.36 ± 0.03 for Model 4 in this study. This was higher than the values of 0.05 to 0.24 reported by Boujenane et al. [20], Boujenane and Diallo [21], and Balasundaram et al. [2] in D’man, Sardi, and Mecheri sheep, respectively. This moderate heritability for W3 indicates that genetic improvement can be achieved through selection for higher body weights in Dağlıç sheep. Meanwhile, Bangar et al. [22] found a higher heritability estimate (0.45) with Model 1 in this trait of Hamali sheep. The maternal heritability and correlation between animal and dam were 0.15 ± 0.02 and −0.24 ± 0.07 in this study. Kiya et al. [23] estimated the direct and maternal heritabilities of 0.21 and 0.16 with a correlation between these effects of −0.45 in Dorper sheep reared in Brazil. The maternal heritability estimate pointed out that W3 is not only dependent on the lamb’s own genetic capacity but also linked to the dam’s milk yield and mothering ability. Similar results were reported by Kushwaha et al. [18] and Sharif et al. [15].

Considering direct and maternal additive effects, Model 3 appeared as the best equation for the average daily gain (ADG). However, some studies [22,24,25,26,27,28,29] suggested other models for this trait in Moghani, Avikalin, Baluchi, Barki, Harnali, and Alpine Merino sheep. The additive heritability estimate for ADG (0.27 ± 0.02) was moderate and consistent with previous values, which ranged from 0.18 to 0.33 [25,29,30,31]. Taskin et al. [32] and Bangar et al. [22] obtained higher additive heritability estimates of 0.61 and 0.43 in Sönmez and Harnali sheep. Maternal heritability of ADG (0.13 ± 0.01) was similar with the earlier reports in Barki and Alpine Merino breeds [28,29]. Several researchers [26,27,31] found lower maternal heritabilities. The differences might be due to the data structure, different breeds, and statistical models used to estimate variance components.

Model 3, comprising direct and maternal genetic effects, was the best fitted one for WW. In agreement with the present study, the results of Bahreini Behzadi et al. [33], Kushwaha et al. [18], Jafaroghli et al. [24], Thiruvenkadan et al. [34], and Mohammadi et al. [35] indicated Model 3 as the most suitable for WW. However, Chauhan et al. [36], Areb et al. [17], Ghaderi-Zefrehei et al. [37], Oyieng et al. [19], Li et al. [29], Sharif et al. [15], Hızlı et al. [16], and Sallam et al. [28] showed different models as the most favorable. The estimate of direct heritability for WW was 0.22 ± 0.02 for Model 3. Higher estimates ranging from 0.31 to 0.50 were observed in different sheep breeds [15,17,19,31,32,38]. The estimate of $h_{a}^{2}$ was consistently within the range of 0.18 to 0.23 reported in the literature [12,16,18,29,33,37] for Dağlıç, Kermani, Chokla, Lori Bakhtiari, Alpine Merino, and Awassi sheep breeds. However, some reports [11,24,35,36] indicated lower heritability estimates in Dağlıç (0.05), Moghani (0.09), Lori (0.006), and Harnali (0.10) sheep. The heritability attributable to maternal effects (0.17) was at the lower boundary of values between 0.17 and 0.28 found in similar studies [15,17,29,33,34,38]. However, Kushwaha et al. [18], Jafaroghli et al. [24], Behrem, [31], Chauhan et al. [36], and Hızlı et al. [16] declared lower $h_{m}^{2}$ estimates in Anatolian Merino, Awassi, Chokla, Harnali, and Moghani sheep. Consistent with our results, Sharif et al. [15] mentioned that the moderate maternal heritability estimates attributed to the lamb’s performance is not only influenced by the lamb’s own genetic potential, but also depends on the dam’s maternal ability.

Model 4, including direct and maternal additive genetic effects along with their covariance, was determined as the best-fitting model for W6 and W12 according to AIC values. Similarly, different researchers reported Model 4 as the best equation for W6 [35] and W12 [27,29]. The heritability estimate of W6 with this model was found to be 0.47 ± 0.05 in Dağlıç sheep. This estimate was higher than those from 0.06 to 0.32 reported of Mecheri [2,34,39], Baluchi [26,27], Lori [35,37], Harnali [22,36], Bonga [17], and Lohi [15] sheep. The direct heritability (0.47 ± 0.05) of W12 was moderately high. Lower estimates were reported for W12 in Moghani [24], Mecheri [2,34,39], Baluchi [27], Lori [37], Harnali [22,36], Bonga [17], Lohi [15], and Dorper [29] sheep within the range of 0.09–0.32. Maternal heritability for W12 (0.26 ± 0.04) was lower than the estimate of 0.39 from Areb et al. [17] in Bonga sheep. The moderate to high heritabilities of the current study indicated that improvement in the post-weaning growth traits in Dağlıç sheep could be achieved by the selection benefitting from these genetic parameters.

Based on the AIC values for each model, Models 3 and 4 stand out for traits we studied in Dağlıç sheep. In other studies, different models provided the best fit to the data. Breed, data set, geography, and the statistical model may produce different results regarding which model is the best fit. Using the most appropriate model in an improvement program provides an accurate estimate of the breeding value of the animals. To acquire successful responses, models that incorporate additive genetic effects, maternal genetic effects, maternal permanent environmental effects, and the direct-maternal covariance will allow additive breeding values to be estimated with improved accuracy.

The genetic, phenotypic, and residual correlations between growth traits estimated with Model 1 are presented in Table 4.

The genetic and phenotypic correlations of BW with the other traits were low to moderate (0.103 ± 0.008–0.431 ± 0.028). If selection is directed to the birth weight, some increases may be expected in dystocia. Similar findings were reported by Sharif et al. [15], and Balasundaram et al. [2] estimated moderate genetic correlations of BW with other growth traits. The strong genetic and phenotypic correlations between W3 and all other traits indicated that this trait could be used as a selection criterion. Jalil-Sarghale et al. [27], Bangar et al. [22], and Balasundaram et al. [2] also observed moderate to high correlations of W3 with other growth traits. This situation indicates that weaning lambs at the age of three months is feasible, taking into account farm conditions and breed characteristics. Furthermore, selection for W3 suggests potential benefits, leading to higher body weights during the weaning and post-weaning stages of the animals. The phenotypic correlation, although smaller than its genetic counterpart, coupled with a low positive residual correlation implies that the genes influencing the traits are similar and environments affecting these traits have a low correlation [40,41]. This scenario is evident in the correlation between W3 and W12. The high genetic correlation, along with a small positive residual correlation accounting for intangible environmental effects, suggests that selection for W3 could effectively enhance W12. In our study, the ADG and WW were strongly correlated with growth traits except for BW. These findings indicate that using the animals with higher ADG and WW in selection improves growth, leading to faster-growing lambs. Similar to our study, Abbasi et al. [26], Mohammadi et al. [35], Areb et al. [17], Behrem [31], and Li et al. [29] reported strong correlations between ADG and WW. However, Haile et al. [42] reported a low genetic correlation (0.23) between ADG and WW in Syrian and Turkish Awassi sheep. The inclusion of dairy breeds of sheep in their analysis may have contributed to this difference in genetic correlations. The genetic (0.995 ± 0.002) and phenotypic (0.939 ± 0.002) correlations between WW and W6 were very high in this study. Mohammadi et al. [35] reported consistent findings for the correlations of WW and W6. Additionally, the correlations between WW and W12 in Dağlıç sheep were greater than the report of Li et al. [29] for Alpine Merino sheep. In the present study, W6 showed strong genetic (0.979 ± 0.006) and phenotypic (0.879 ± 0.003) correlations with W12. Jalil-Sarghale et al. [27] and Balasundaram et al. [2] found similar results for W6 and W12 in Baluchi and Mecheri sheep. These results imply that the animals heavier at weaning would have greater live weight at one year of age. The reason behind the genetic and phenotypic correlations observed in this study may be the similarity of genes responsible for growth during different stages of the life cycle. The strong correlations observed among different variables can be attributed to the influence of part-whole relationships between traits. This is because weights at later ages rely on earlier weights, causing an increase in correlations over time. As time progresses, the increasing dependency of later weight measures on preceding values contributes to the high correlations. Fischer et al. [43] and Atoui et al. [44] also reported the similar pattern in Poll Dorset sheep and Tunisian goat. As a result, selection based on any of these body weights will result in positive responses in other traits because antagonistic genetic correlations were not observed between the traits.

## 4. Conclusions

The results of this study indicate that the direct additive genetic and maternal effects play a crucial role in the phenotypic variation in pre- and post-weaning growth traits of lambs. The moderate to large heritabilities reported in our study indicate that selection for growth traits should be successful in the Dağlıç breed of sheep. Different models incorporating direct additive genetic, maternal genetic, and maternal permanent environmental effects, as well as the correlation between the direct and maternal genetic effects, should be taken into consideration in breeding programs. Determining the best-fitting model for any economically important trait provides an increase in the accuracy of estimates for additive genetic breeding values and will increase the rate of genetic improvement. The strong positive genetic correlations among weaning and post-weaning growth traits and genetic variances should be considered in the selection program to improve growth traits in this breed. Further investigations would be beneficial in identifying the genes that account for genetic variation in growth rate.

## Figures and Tables

**Table 1 animals-14-00108-t001:** Data structure for pre- and post-weaning growth traits of native Dağlıç sheep.

	Traits					
Items	BW	W3	ADG	WW	W6	W12
No. of records	21,735	17,679	20,229	20,229	6340	5261
No. of dams	7464	6587	7167	7167	3248	2660
No. of dams with progeny	7446	6540	7145	7145	3154	2569
No. of dams with progeny and records	4084	3263	3902	3902	1595	1225
Average number of progeny per dam	3.47	3.16	3.37	3.37	1.84	1.78
No. of sires	270	267	270	270	257	239
No. of sires with progeny	270	267	270	270	257	239
No. of sires with progeny and records	187	163	183	183	158	143
Average number of progeny per sire	57.85	52.06	55.69	55.69	18.25	16.42
Mean	3.38	17.46	161.11	23.39	26.96	32.51
Standard Deviation	0.62	4.49	45.86	6.01	5.45	6.68

Abbreviations: BW, birth weight; W3, weight at three months of age; ADG, average daily weight gain; WW, weight at weaning (avg. 124.89 days); W6, weight at six months of age; W12, weight at 12 months of age. The unit of all traits is kg, except for ADG (g).

**Table 2 animals-14-00108-t002:** Variance components and genetic parameters for BW, W3, and ADG with LogL and AIC values for different models.

Traits	Model	$\sigma_{a}^{2}$	$\sigma_{pe}^{2}$	$\sigma_{m}^{2}$	$\sigma_{am}$	$\sigma_{e}^{2}$	$\sigma_{P}^{2}$	$h_{a}^{2}$	${pe}^{2}$	$h_{m}^{2}$	$r_{am}$	$h_{T}^{2}$	LogL	AIC
BW	1	0.16 ± 0.01				0.20 ± 0.01	0.36 ± 0.00	0.45 ± 0.02				0.45 ± 0.02	1030.775	−2057.55
	2	0.15 ± 0.01	0.01 ± 0.00			0.19 ± 0.01	0.36 ± 0.00	0.42 ± 0.02	0.04 ± 0.01			0.42 ± 0.02	1044.812	−2083.624
	3	0.13 ± 0.01		0.03 ± 0.00		0.20 ± 0.01	0.36 ± 0.00	0.35 ± 0.02		0.08 ± 0.01		0.39 ± 0.02	1078.131	**−2150.262**
	4 *													
	5	0.12 ± 0.01	0.00 ± 0.00	0.04 ± 0.01		0.21 ± 0.01	0.37 ± 0.00	0.33 ± 0.02	0.00 ± 0.01	0.11 ± 0.02		0.38 ± 0.02	1071.032	−2134.064
	6 *													
W3	1	7.36 ± 0.34				9.66 ± 0.28	17.03 ± 0.20	0.43 ± 0.02				0.43 ± 0.02	−33,311.711	66,627.422
	2	6.48 ± 0.36	1.23 ± 0.16			9.31 ± 0.28	17.02 ± 0.20	0.38 ± 0.02	0.07 ± 0.01			0.38 ± 0.02	−33,282.258	66,570.516
	3	5.33 ± 0.38		1.95 ± 0.19		9.78 ± 0.27	17.06 ± 0.20	0.31 ± 0.02		0.11 ± 0.01		0.37 ± 0.02	−33,242.251	66,490.502
	4	6.06 ± 0.52		2.50 ± 0.29	−0.92 ± 0.32	9.35 ± 0.35	16.99 ± 0.20	0.36 ± 0.03		0.15 ± 0.02	−0.24 ± 0.07	0.35 ± 0.02	−33,237.381	**66,482.762**
	5	5.33 ± 0.39	0.00 ± 0.24	1.95 ± 0.29		9.78 ± 0.29	17.07 ± 0.20	0.31 ± 0.02	0.00 ± 0.01	0.11 ± 0.02		0.37 ± 0.02	−33,242.256	66,492.512
	6	6.06 ± 0.52	0.00 ± 0.24	2.50 ± 0.38	−0.92 ± 0.32	9.35 ± 0.36	16.99 ± 0.20	0.36 ± 0.03	0.00 ± 0.01	0.15 ± 0.02	−0.24 ± 0.07	0.35 ± 0.02	−33,237.387	66,484.774
ADG	1	747.53 ± 33.02				1014.92 ± 26.58	1762.45 ± 19.10	0.42 ± 0.02				0.42 ± 0.02	−84,994.784	169,993.568
	2	651.20 ± 35.74	116.77 ± 15.43			991.39 ± 26.87	1759.36 ± 19.15	0.37 ± 0.02	0.07 ± 0.01			0.37 ± 0.02	−84,966.485	169,938.97
	3	472.31 ± 37.46		221.42 ± 18.76		1066.03 ± 26.80	1759.76 ± 19.33	0.27 ± 0.02		0.13 ± 0.01		0.33 ± 0.02	−84,907.774	**169,821.548**
	4	473.98 ± 45.05		222.70 ± 25.89	−2.16 ± 27.44	1065.06 ± 31.37	1759.58 ± 19.35	0.27 ± 0.03		0.13 ± 0.02	−0.01 ± 0.08	0.33 ± 0.02	−84,907.897	169,823.794
	5	472.37 ± 38.09	0.00 ± 22.36	221.38 ± 28.02		1066.01 ± 28.18	1759.76 ± 19.33	0.27 ± 0.02	0.00 ± 0.01	0.13 ± 0.02		0.33 ± 0.02	−84,907.774	169,823.548
	6	473.96 ± 45.44	0.00 ± 22.40	222.70 ± 33.69	−2.15 ± 27.46	1065.07 ± 32.39	1759.58 ± 19.35	0.27 ± 0.03	0.00 ± 0.01	0.13 ± 0.02	−0.01 ± 0.08	0.33 ± 0.02	−84,907.898	169,825.796

Abbreviations: $\sigma_{a}^{2}$, additive genetic variance; $\sigma_{pe}^{2}$, permanent environmental variance of dam; $\sigma_{m}^{2}$, maternal genetic variance; $\sigma_{am}$, covariance between additive and maternal genetic effect; $\sigma_{e}^{2}$, residual variance; $\sigma_{P}^{2}$, phenotypic variance; $h_{a}^{2}$, direct heritability; ${pe}^{2}$, ratio of variance due to maternal permanent environmental effect to total phenotypic variance; $h_{m}^{2}$, maternal heritability; $r_{am}$, correlation between additive and maternal genetic effect; $h_{T}^{2}$, total heritability; BW, birth weight; W3, weight at three months of age; ADG, average daily weight gain. *: Convergence was not reached with AIREML. The best fitted model according to AIC is shown in bold type.

**Table 3 animals-14-00108-t003:** Variance components and genetic parameters for WW, W6, and W12 with LogL and AIC values for different models.

Traits	Model	$\sigma_{a}^{2}$	$\sigma_{pe}^{2}$	$\sigma_{m}^{2}$	$\sigma_{am}$	$\sigma_{e}^{2}$	$\sigma_{P}^{2}$	$h_{a}^{2}$	${pe}^{2}$	$h_{m}^{2}$	$r_{am}$	$h_{T}^{2}$	LogL	AIC
WW	1	11.42 ± 0.49				14.90 ± 0.39	26.32 ± 0.29	0.43 ± 0.02				0.43 ± 0.02	−42,482.472	84,968.944
	2	9.10 ± 0.54	2.52 ± 0.24			14.59 ± 0.40	26.21 ± 0.29	0.35 ± 0.02	0.10 ± 0.01			0.35 ± 0.02	−42,424.519	84,855.038
	3	5.84 ± 0.53		4.39 ± 0.29		16.04 ± 0.39	26.27 ± 0.29	0.22 ± 0.02		0.17 ± 0.01		0.31 ± 0.02	−42,342.115	**84,690.23**
	4	5.62 ± 0.59		4.10 ± 0.40	0.40 ± 0.39	16.19 ± 0.43	26.31 ± 0.29	0.21 ± 0.02		0.16 ± 0.02	0.08 ± 0.09	0.31 ± 0.02	−42,341.753	84,691.506
	5	5.84 ± 0.54	0.00 ± 0.35	4.39 ± 0.45		16.04 ± 0.40	26.28 ± 0.29	0.22 ± 0.02	0.00 ± 0.01	0.17 ± 0.02		0.31 ± 0.02	−42,342.123	84,692.246
	6	5.62 ± 0.59	0.00 ± 0.35	4.10 ± 0.54	0.40 ± 0.39	16.19 ± 0.44	26.31 ± 0.29	0.21 ± 0.02	0.00 ± 0.01	0.16 ± 0.02	0.08 ± 0.09	0.31 ± 0.02	−42,341.761	84,693.522
W6	1	11.76 ± 0.64				7.84 ± 0.48	19.59 ± 0.38	0.60 ± 0.03				0.60 ± 0.03	−12,320.049	24,644.098
	2	10.90 ± 0.67	2.36 ± 0.41			6.35 ± 0.52	19.61 ± 0.38	0.56 ± 0.03	0.12 ± 0.02			0.56 ± 0.03	−12,305.355	24,616.71
	3	8.27 ± 0.73		4.32 ± 0.48		7.11 ± 0.49	19.69 ± 0.39	0.42 ± 0.04		0.22 ± 0.02		0.53 ± 0.03	−12,274.707	24,555.414
	4	9.29 ± 1.03		5.06 ± 0.66	−1.14 ± 0.68	6.41 ± 0.69	19.62 ± 0.39	0.47 ± 0.05		0.26 ± 0.03	−0.17 ± 0.09	0.52 ± 0.03	−12,273.264	**24,554.528**
	5	8.27 ± 0.77	0.00 ± 0.58	4.32 ± 0.72		7.11 ± 0.56	19.70 ± 0.39	0.42 ± 0.04	0.00 ± 0.03	0.22 ± 0.04		0.53 ± 0.03	−12,274.71	24,557.42
	6	9.29 ± 1.04	0.00 ± 0.61	5.07 ± 0.94	−1.14 ± 0.70	6.41 ± 0.71	19.63 ± 0.39	0.47 ± 0.05	0.00 ± 0.03	0.26 ± 0.05	−0.17 ± 0.09	0.52 ± 0.03	−12,273.268	24,556.536
W12	1	14.02 ± 0.98				14.14 ± 0.80	28.16 ± 0.58	0.50 ± 0.03				0.50 ± 0.03	−11,257.332	22,518.664
	2	13.03 ± 1.03	3.18 ± 0.67			11.99 ± 0.88	28.20 ± 0.59	0.46 ± 0.03	0.11 ± 0.02			0.46 ± 0.03	−11,246.6	22,499.2
	3	10.84 ± 1.10		5.43 ± 0.73		12.16 ± 0.80	28.42 ± 0.60	0.38 ± 0.04		0.19 ± 0.03		0.48 ± 0.03	−11,224.35	22,454.7
	4	13.21 ± 1.60		7.20 ± 1.08	−2.74 ± 1.12	10.59 ± 1.11	28.26 ± 0.60	0.47 ± 0.05		0.26 ± 0.04	−0.28 ± 0.09	0.45 ± 0.03	−11,221.275	**22,450.55**
	5	10.83 ± 1.13	0.00 ± 1.00	5.43 ± 1.13		12.17 ± 0.90	28.43 ± 0.60	0.38 ± 0.04	0.00 ± 0.04	0.19 ± 0.04		0.48 ± 0.03	−11,224.352	22,456.704
	6	13.21 ± 1.60	0.00 ± 1.05	7.20 ± 1.54	−2.74 ± 1.15	10.59 ± 1.14	28.27 ± 0.60	0.47 ± 0.05	0.00 ± 0.04	0.26 ± 0.05	−0.28 ± 0.09	0.45 ± 0.03	−11,221.278	22,452.556

Abbreviations: $\sigma_{a}^{2}$, additive genetic variance; $\sigma_{pe}^{2}$, permanent environmental variance of dam; $\sigma_{m}^{2}$, maternal genetic variance; $\sigma_{am}$, covariance between additive and maternal genetic effect; $\sigma_{e}^{2}$, residual variance; $\sigma_{P}^{2}$, phenotypic variance; $h_{a}^{2}$, direct heritability; ${pe}^{2}$, ratio of variance due to maternal permanent environmental effect to total phenotypic variance; $h_{m}^{2}$, maternal heritability; $r_{am}$, correlation between additive and maternal genetic effect; $h_{T}^{2}$, total heritability; WW, weight at weaning; W6, weight at six months of age; W12, weight at 12 months of age. The best fitted model according to AIC is shown in bold type.

**Table 4 animals-14-00108-t004:** Genetic (below diagonal), phenotypic, and residual correlations (above diagonal) for pre- and post-weaning growth traits.

	BW	W3	ADG	WW	W6	W12
BW	-	0.240 ± 0.008 (0.079 ± 0.021)	0.103 ± 0.008 (−0.059 ± 0.019)	0.220 ± 0.007 (0.089 ± 0.019)	0.201 ± 0.013 (0.038 ± 0.038)	0.166 ± 0.014 (−0.025 ± 0.039)
W3	0.431 ± 0.028	-	0.910 ± 0.001 (0.871 ± 0.004)	0.893 ± 0.002 (0.848 ± 0.005)	0.832 ± 0.004 (0.689 ± 0.020)	0.638 ± 0.009 (0.365 ± 0.036)
ADG	0.317 ± 0.030	0.965 ± 0.004	-	0.956 ± 0.001 (0.938 ± 0.002)	0.909 ± 0.002 (0.818 ± 0.012)	0.709 ± 0.007 (0.422 ± 0.032)
WW	0.391 ± 0.028	0.958 ± 0.004	0.979 ± 0.002	-	0.939 ± 0.002 (0.862 ± 0.008)	0.717 ± 0.007 (0.413 ± 0.032)
W6	0.362 ± 0.043	0.939 ± 0.009	0.975 ± 0.004	0.995 ± 0.002	-	0.879 ± 0.003 (0.760 ± 0.016)
W12	0.394 ± 0.052	0.889 ± 0.020	0.940 ± 0.013	0.963 ± 0.012	0.979 ± 0.006	-

Residual correlations in parentheses. Abbreviations: BW, birth weight; W3, weight at three months of age; ADG, average daily weight gain; WW, weight at weaning; W6, weight at six months of age; W12, weight at 12 months of age.

## Data Availability

The data sets used in this study belong to the General Directorate of Agricultural Research and Policies, Ministry of Agriculture and Forestry, Türkiye. The data sets of the current study will be made available on reasonable request with permission from the related government agency.

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
