# Peer review of "Estimation of (Co) Variance Components and Genetic Parameters for Pre- and Post-Weaning Growth Traits in Dağlıç Sheep"

_animals, 2023, doi:10.3390/ani14010108_

Round 1
Reviewer 1 Report
Comments and Suggestions for Authors
While the paper provides valuable insights into (co)variance components and genetic parameters for pre- and post-weaning growth traits in Dağlıç sheep, there are areas for improvement:
The paper mentions that Model 3 was found to be the most suitable for certain traits, and Model 4 for others, but there's a lack of clarity in explaining why these models were considered the most appropriate. Providing a more detailed rationale or comparison with other models would enhance the understanding of the reader.
The paper briefly mentions the use of the Average Information-Restricted Maximum Likelihood algorithm but lacks details on why this algorithm was chosen over others, and how its application may have influenced the results. Including this information would strengthen the methodological rigor of the study.
The discussion section could benefit from a more in-depth analysis of the obtained results. For example, exploring potential reasons behind the observed genetic and phenotypic correlations, and discussing how these findings align with or differ from existing literature would add depth to the interpretation.
Certainly, if the introduction is deemed too brief and the materials and methods section lacks detail, consider expanding them to provide a more comprehensive context for your study and a clearer understanding of your research approach.
- Explain the methods used for collecting data on pre- and post-weaning growth traits.
- Clarify how birth weight, average daily gain, and other traits were measured or recorded.
-
- Provide more information on how the Average Information-Restricted Maximum Likelihood algorithm was applied.
- Discuss any statistical considerations or potential limitations in your approach.
Author Response
REVIEWER 1
Dear Editor/s and Reviewers, we are thankful to you for your kindly interest and time to review and making suggestion to our manuscript. Comments, questions and suggestion of the reviewers were very useful to improve our manuscript. Reviewers’ suggestions and responses are presented as per the Reviewer.
- The paper mentions that Model 3 was found to be the most suitable for certain traits, and Model 4 for others, but there's a lack of clarity in explaining why these models were considered the most appropriate. Providing a more detailed rationale or comparison with other models would enhance the understanding of the reader.
As we stated in lines 105-110, we performed a log-likelihood test based on "Akaike's information criterion" to determine which model is most suitable for identifying variance components in the relevant trait. As seen in the AIC column in the table 2-3, the model with the lowest AIC value was considered the most suitable model.
- The paper briefly mentions the use of the Average Information-Restricted Maximum Likelihood algorithm but lacks details on why this algorithm was chosen over others, and how its application may have influenced the results. Including this information would strengthen the methodological rigor of the study.
WOMBAT program was used to find variance components of the mentioned traits. This program uses the AI-REML algorithm by default developed by Thompson et al. (doi:10.1098/rstb.2005.1676). This algorithm provides reliable convergence for the analyses even for difficult problems involving numerous traits or multiple random effects (Meyer, 2007, doi: 10.1631%Fjzus.2007.B0815).
- The discussion section could benefit from a more in-depth analysis of the obtained results. For example, exploring potential reasons behind the observed genetic and phenotypic correlations, and discussing how these findings align with or differ from existing literature would add depth to the interpretation.
According to Reviewers’ suggestion, Discussion related to the correlations have improved and additions were highlighted.
- Certainly, if the introduction is deemed too brief and the materials and methods section lacks detail, consider expanding them to provide a more comprehensive context for your study and a clearer understanding of your research approach.
We consider that the introduction is average length and has a sufficient background. The other two reviewers agreed with us and noted that the introduction provided sufficient background and included all relevant references.
- Explain the methods used for collecting data on pre- and post-weaning growth traits.
In the scope of “Community-based animal improvement program” project coordinated by the General Directorate of the Agricultural Research and Policies of the Ministry of Agriculture and Forestry of Türkiye, a technical staff and research team collected the weight records from the farms. So, the data used in this research were comprised of these weight records. Also, explaining for pre- and post-weaning growth traits was presented in Materials and Method section as shown below,
“The lambs were weighed within 24 hours after birth and raised with their mothers until weaning. Average weaning age was 124.89 days in this research. The weights of the lambs were taken consecutively until the lambs were older than one year. The W3, W6 and W12 were estimated with interpolation using the weight records. The ADG was calculated from birth to weaning.”
- Clarify how birth weight, average daily gain, and other traits were measured or recorded.
Information on how birth weight, ADG, and other traits are measured or calculated is clarified in lines 69-73.
- Provide more information on how the Average Information-Restricted Maximum Likelihood algorithm was applied.
WOMBAT program uses the AI-REML algorithm by default to provides reliable convergence for the analyses even for difficult problems involving numerous traits or multiple random effects. As is seen in lines 75-77.
- Discuss any statistical considerations or potential limitations in your approach.
We did not conduct a study to evaluate the AI-REML algorithm. We used the WOMBAT program, which uses this algorithm to estimate variance components. Therefore, we did not have this discussion as we thought that discussing statistical considerations or potential limitations of this algorithm was outside the scope of the research.

Reviewer 2 Report
Comments and Suggestions for Authors
Estimation of (co)variance components and genetic parameters 2 for pre- and post-weaning growth traits in Dağlıç sheep
Keywords: Variance component; genetic parameters; heritability; maternal effect; growth traits; 29 Dağlıç sheep
There are words in the title as keywords. It should not be repeated. Choose one of the two places to place it and replace it in the other.
In line 111: The total heritability, change the H for h: ( h2T) for each model was calculated as h2T = ….. Change the table 2, too.
Table 1, needs to inform the age of weaning weight [; WW, weight at weaning;]
In table 3, the authors could discuss why in models 1 and 2 the direct additive heritability values are equal to the total heritability values. However, doesn't this happen for other models?
The authors could simulate responses in a herd when using each of the models and show the possible answers.
Make a discussion more about the models; the answers and in what situation to use them. Should you use it or not? It's because?
Do heritability estimates using the different models impact the breeding program? Or does it matter if you use any of the estimates? Authors must show the applicability of this.
In lines 96 to 103: the authors built two models, 5 and 6 the most complete. However, I miss a discussion because these models were not as efficient as 4? Do you consider reporting their results and why they were not chosen as better models?
“It 96 was assumed that direct additive genetic, maternal additive genetic, maternal permanent 97 environmental, and residual effects are normally and independently distributed with 98 mean 0 and variance 𝑨𝜎2𝑎, 𝑨𝜎2m, 𝑰𝒅𝜎2pe, 𝑰𝒏𝜎2pe and 𝑰𝒏𝜎2e respectively. ….. 𝑰𝒅 and 𝑰𝒏 are the identity matrices with orders equal to the number of dams and 103 number of lambs. “
For example: the authors chose the best model for WW, 4, based on the AIC. However, this model failed to estimate the permanent environment effect 𝜎2pe . Could this not create a bias in the heritability estimate?
The conclusion is not a conclusion: The authors show an objective (Not a conclusion). The other information would fit well into discussions. But these are not conclusions. Authors must make simple and direct conclusions suggesting actions that can be carried out based on what was developed in the paper (work)
Author Response
REVIEWER 2
Dear Editor/s and Reviewers, we are thankful to you for your kindly interest and time to review and making suggestion to our manuscript. Comments, questions and suggestion of the reviewers were very useful to improve our manuscript. Reviewers’ suggestions and responses are presented as per the Reviewer.
- There are words in the title as keywords. It should not be repeated. Choose one of the two places to place it and replace it in the other.
We have re-handled the keywords.
- In line 111: The total heritability, change the H for h: ( h) foreach model was calculated as h = ….. Change the table 2, too.
As suggested by the reviewer, “H” were converted to “h” in everywere of the manuscripts.
- Table 1, needs to inform the age of weaning weight [; WW, weight at weaning;]
Average weaning age in this study were inserted into the Materials and Method section and Table 1.
- In table 3, the authors could discuss why in models 1 and 2 the direct additive heritability values are equal to the total heritability values. However, doesn't this happen for other models?
In lines 110-111, we described the formula of total heritability calculation. Why the reason of the direct additive heritability equal to total heritability in model 1 and 2 is the only additive genetic effect of the animal included in model 1 and 2. However, in the other models, the effects of maternal genetic of animal and covariance between animal and dam were considered as random effects.
- The authors could simulate responses in a herd when using each of the models and show the possible answers. Make a discussion more about the models; the answers and in what situation to use them. Should you use it or not? It's because? Do heritability estimates using the different models impact the breeding program? Or does it matter if you use any of the estimates? Authors must show the applicability of this.
A paragraph (line 221-228) containing the answers to the questions mentioned here has been added to the relevant discussion section.
- In lines 96 to 103: the authors built two models, 5 and 6 the most complete. However, I miss a discussion because these models were not as efficient as 4? Do you consider reporting their results and why they were not chosen as better models?
We revealed that Model 4 is the best equation for some traits based on AIC values derived from log-likelihood values, but it should not be overlooked that this model may vary depending on some factors such as the breed, herd, data structure and selection. In studies using another data set or breed, Model 5 or 6 may also appear as the best model. Here, the log-likelihood values which showed the model unbiasedly estimate the variance components are used to determine the most appropriate model.
- For example: the authors chose the best model for WW, 4, based on the AIC. However, this model failed to estimate the permanent environment effect ?2pe . Could this not create a bias in the heritability estimate?
In weaning weight, Model 4 is the best model based on AIC. Here, AIREML has converged the model without considering the permanent environmental effect. Additive genetic effect, maternal genetic effect, and correlation between offspring and dam were considered in Model 4. This model did not fail to estimate the permanent environmental effect, model 4 did not contain this effect.
- The conclusion is not a conclusion: The authors show an objective (Not a conclusion). The other information would fit well into discussions. But these are not conclusions. Authors must make simple and direct conclusions suggesting actions that can be carried out based on what was developed in the paper (work).
According to reviewers’ suggestions, we re-handled the conclusion section.

Reviewer 3 Report
Comments and Suggestions for Authors
I have read your submission with high interest. I agree with the overall idea that sustainable improvement in animal/ sheep production is not possible without improvement in breeding and application of state-of-the art technology to achieve genetic progress and increase production capacity of this sector of agriculture to feed people.
With this I was going through your submission and have to state that the manuscript is well written, with clear presentation of current state of the art, material(data) used in analysis as well as statistical procedures.
I would like to stress, thatr I like this "classical" way, authors describe methodology, with models and variables definition.
There are no major concerns regarding the Results and Discussion.
However, I was wondering about the information present in Table 1 when looking at average number of progeny per dam. Basic statistical rules says that there is minimum design of 3 observation per group, i.e. it means that you are hardly reaching this limit when looking on average group size in the models.
I have not find any further information in the Discussion section and therefore ask authors to check what is the minimum number of observation per class in the models because this can significantly affect inter and intra class variation and thus results. However (overal) phenotypic variance is relatively high.
As the coeff.variance is easy to calculate from mean and SD, I would recommend presence of either SD or CV, not both in Table 1.
Regarding the later Table 4. I would recommend to present Genetic and environmental correlations instead of phenotypic to allow check for correlation matrix soundness.
However, there are no further major nor minor issues to be considered.
Author Response
REVIEWER 3
Dear Editor/s and Reviewers, we are thankful to you for your kindly interest and time to review and making suggestion to our manuscript. Comments, questions and suggestion of the reviewers were very useful to improve our manuscript. Reviewers’ suggestions and responses are presented as per the Reviewer.
- However, I was wondering about the information present in Table1 when looking at average number of progeny per dam. Basic statistical rules says that there is minimum design of 3 observation per group, i.e. it means that you are hardly reaching this limit when looking on average group size in the models.
Selection is already being applied in these herds. For rapid genetic progress, sheep are reformed from the herd after the age of 4 or 5, and young animals are involved in the herd instead. The herd replacement rate is about %25 in the herds. Dağlıç sheep is not a prolific breed. This is the reason for observation size per dam.
- I have not find any further information in the Discussion section and therefore ask authors to check what is the minimum number of observation per class in the models because this can significantly affect inter and intra class variation and thus results. However (overal) phenotypic variance is relatively high.
The average number of offspring per dam is lower in many studies. For example, (https://doi.org/10.3390/ani13030454). To the best of our knowledge, the data which was used in the analyses has a strong data structure.
- As the coeff.variance is easy to calculate from mean and SD, I would recommend presence of either SD or CV, not both in Table1.
CV parameters were removed from the Table 1.
- Regarding the later Table 4. I would recommend to present Genetic and environmental correlations instead of phenotypic to allow check for correlation matrix soundness.
The reader can infer an interaction with the word environmental. Here, we wanted to reveal the genetic and phenotypic correlations between traits. Since environmental effects are included in the model, Genetic and Phenotypic would be more suitable for us here.
Round 2
Reviewer 3 Report
Comments and Suggestions for Authors
Sorry, not sufficient response. You dint undergo any further reading to reply with qualified response.
Removing CV form tables is not enough. NOBs per class clearly states whether your model make sense or not. Regarding your reply it means that your model is accounting for non-significant effects.
Presentation of full correlation matrix could confirm that your model converged. Otherwise (hiding values) one would state that your results are not calculated reliably.
Author Response
REVIEWER 3
Dear Editor/s and Reviewers, we are thankful to you for your kindly interest and time to review and making suggestion to our manuscript. Comments, questions and suggestion of the reviewers were very useful to improve our manuscript. Reviewers’ suggestions and responses are presented as per the Reviewer.
- I have not find any further information in the Discussion section and therefore ask authors to check what is the minimum number of observation per class in the models because this can significantly affect inter and intra class variation and thus results. However (overal) phenotypic variance is relatively high.
NOBs per class clearly states whether your model make sense or not.
First of all, I would like to point out that we cannot understand you clearly and in detail. I think the high phenotypic variance makes you consider that there are inadequacies in the number of subgroups in environmental factors whose effects can be eliminated. The minimum number of observations per class is at the 12th month body weight with 112 observations. Besides, the number of animals is quite high in other traits. Although phenotypic variances are quite high, improvement studies have only recently started in this population. In fact, raw variances are even higher than phenotypic variances. Raw variances can be presented in Table 1 if desired. In the materials and methods section, we stated that we also included fixed effects in the model, which were determined to be important after least squares analysis. However, if there are issues, we do not understand and comprehend, we kindly want you to inform us more clearly to improve our manuscript. Regards
- Regarding the later Table 4. I would recommend to present Genetic and environmental correlations instead of phenotypic to allow check for correlation matrix soundness.
Presentation of full correlation matrix could confirm that your model converged. Otherwise (hiding values) one would state that your results are not calculated reliably.
We can share all the values of our correlation calculations. We have no intention of hiding the findings. We did not calculate environmental correlations. But we added the residual correlations in brackets to the table 4. Also, we made some statements in discussion section about correlations. However, if you think that we do not fully understand you, we would be grateful if you give us feedback so that we can implement your valuable suggestions.

Round 3
Reviewer 3 Report
Comments and Suggestions for Authors
I am not co-author and therefore not feeling qualified to further develop the paper.
However when looking on table 1. , row 5 (Average No.of progenies per dam) W6, W12 .... 1.84 and 1.78, respectively than it means there were from 1 up to 3 observation per group (dam)!
What is the average of 1 observation i.e. variation or SD? If average no. is under 2 than it means that there is excess of max.2 obs per group....and again what sense it makes calculate average and variation from 2 observations ..... in sense of estimation of variation components which is based on comparison of intra and inter class variation?
Comments on the Quality of English LanguageN/A
Author Response
REVIEWER 3
Dear Editor/s and Reviewers, we are thankful to you for your kindly interest and time to review and making suggestion to our manuscript. Comments, questions and suggestion of the reviewers were very useful to improve our manuscript. Reviewers’ suggestions and responses are presented as per the Reviewer.
1-) I am not co-author and therefore not feeling qualified to further develop the paper.
However when looking on table 1. , row 5 (Average No.of progenies per dam) W6, W12 .... 1.84 and 1.78, respectively than it means there were from 1 up to 3 observation per group (dam)!
What is the average of 1 observation i.e. variation or SD? If average no. is under 2 than it means that there is excess of max.2 obs per group....and again what sense it makes calculate average and variation from 2 observations ..... in sense of estimation of variation components which is based on comparison of intra and inter class variation?
In this study, REML technic was used to solve the problem of animals with a single observation in estimating genetic parameters. This technic is known for its ability to handle datasets with a relatively small number of observations per individual. The animal model analysis with REML using information from all known relationships among animals has been used to estimate genetic parameters for this study. The relevant information was added to the Materials and Method section (line 73-74).